# N, S, O Self-Doped Carbon Derived from Grapefruit Peel for High-Performance Supercapacitors

**DOI:** 10.3390/ma16134577

**Published:** 2023-06-25

**Authors:** Yi Wang, Liangqun Wang, Xihong Lu

**Affiliations:** 1College of Chemistry and Material Engineering, Guiyang University, Guiyang 550005, China; wy742011@hotmail.com; 2Guizhou Xifeng Phosphate Mine Co., Ltd., Xifeng 551100, China; 3The Key Laboratory of Low-Carbon Chem & Energy Conservation of Guangdong Province, School of Chemistry, Sun Yat-Sen University, Guangzhou 510275, China

**Keywords:** self-doped, carbon, grapefruit peel, supercapacitors

## Abstract

The development of high-capacity carbon for supercapacitors is highly desirable but challenging. In this work, we design a N, S, O self-doped carbon electrode (NSOC-800) with high capacitance and good stability via the carbonization of grapefruit peel via a one-step KOH activation method without extra dopants. The existence of heteroatoms enables the NSOC-800 to have a high specific capacitance of 280 F/g and a great cycling performance, with 90.1% capacitance retention after 5000 cycles. Moreover, the symmetric supercapacitor with NSOC-800 electrodes delivers a maximum energy density of 5 Wh/kg with a power density of 473 W/kg. Such a promising method to achieve carbon materials with self-doping heteroatwoms is of great significance for developing highly efficient electrodes for energy storage devices.

## 1. Introduction

The increasing development of economics and the rapid growth of population accelerate the development of sustainable, clean, and green energy storage technologies and devices [1,2,3,4,5]. Among the various energy storage devices, lithium ion batteries with a high-energy density and a reversible charging/discharging process have dominated the energy market over the past decades [6,7,8]. However, the further advancement of lithium ion batteries is impeded by the limited lithium resources and safety problems due to flammable organic electrolytes [9,10,11,12]. In this regard, supercapacitor devices have attracted overwhelming attention by virtue of cost-efficiency, environmental compatibility, and having secure and efficient charge/discharge processes with cycling stability, which can be divided into two types of reaction mechanism, i.e., electric double-layer capacitors (EDLC) and pseudo-capacitors [13,14,15,16,17,18]. Thus, the electrochemical properties of supercapacitor devices are determined by their reaction mechanisms. Carbon-based materials are identified as the most widely used electrode material for supercapacitor devices, which not only applies to electrochemical double-layer capacitors, but also to the pseudo-capacitors [19,20,21,22,23]. Regretfully, the currently comprehensive performance of carbon-based materials is far from the requirement for wide commercial applications of supercapacitor devices owing to inferior electronic conductivity and practical capacitance.

In terms of the issues, many prevailing strategies have been dedicated to achieving excellent electrochemical performances of carbon-based materials such as structural design, which is favorable for rapid ion diffusion, as well as element doping, which can improve the ion transport efficiency [24,25,26,27,28,29,30,31], enhance the surface hydrophilicity [32,33,34,35] and provide additional faradaic capacitance [36,37,38,39,40]. A case in point is the study by Hou et al., who developed a replicating and embossing strategy to prepare self-supported three-dimensional (3D) porous reduced graphene oxide (rGO) film for electrochemical performance [41]. Benefiting from the unique 3D porous structure, the obtained rGO film in a H_2_SO_4_ electrolyte delivered a large specific capacitance of 206 F/g at a current density of 1 A/g and an outstanding cycling stability of 64% capacitance retention after 5000 charge/discharge cycles. Moreover, a N/S co-doped three dimensional (3D) graphene hydrogel (N/S-3DGH) was prepared by a facile hydrothermal treatment using ammonia and thiourea as source of nitrogen (N) and sulfur (S) for supercapacitor electrode application. The introduction of heteroatoms into graphene endows the carbon-based electrode with an increased activation region on the graphene surface, thus enhancing the energy storage ability [42].

Notwithstanding the great progress made so far, the comprehensive properties of carbon-based materials is still far from satisfactory, which has resulted in complicated processes, costs for extra dopants and so on. Therefore, in this work, we developed a straightforward one-step KOH activation method without extra dopants to prepare a N, S, O self-doped carbon derived from grapefruit peel (NSOC-800) for high-performance supercapacitors. The N, S, O self-doping enables the carbon material to have a high electronic conductivity and fast ion diffusion rate, in turn improving the electrochemical performance of the electrode material. Stimulated by this distinct merit, the symmetrical supercapacitor device assembled with NSOC-800 electrodes in an alkaline electrolyte delivers a high specific capacitance of 75 F/g and a good cycle stability with only 0.003% capacitance attenuation of each cycle per 5000 cycles on average.

## 2. Materials and Methods

### 2.1. Materials and Reagents

All chemicals are analytically pure, meaning that they can be used directly without any purification treatment; grapefruit peel (from Sinopharm Chemical Regent Co., Ltd., Beijing, China), potassium hydroxide (KOH) (from Macklin Biochemical Co., Ltd., 95%, Shanghai, China) and Ni foam (thickness of 1 mm, Taiyuan Power Source Battery Co., Ltd., Taiyuan, China).

### 2.2. Preparation of NSOC-800 Electrode

The NSOC-800 sample was synthesized by a straightforward one-step approach. Specifically, 1 g grapefruit peel powder and 4 g KOH were added to a porcelain boat after homogeneous mixing, followed by annealing at 873 K for 2 h under N_2_ flow in a tube furnace. After carbonization, the black powder was washed with deionized water and dried at 80 °C for 10 h, and the NSOC-800 was finally prepared. To prepare the NSOC-800 electrode, the powder, polyvinylidene fluoride (PVDF) and acetylene black (the weight ratio of 8:1:1) were dispersed into 1-methyl-2-pyrrolidinone (NMP). The obtained slurry was coated on the Ni foam after magnetically stirring for 5 h, and dried in a vacuum condition at 60 °C for one night. The mass loading of the NSOC-800 electrode was 4 mg cm^−2^. The aqueous alkaline symmetric supercapacitor was assembled with the NSOC-800 electrodes and 6 M KOH, in which the area of electrodes is 1 cm^2^ (denoted as NSOC-800//NSOC-800).

### 2.3. Characterization

The morphology of the NSOC-800 was examined by scanning electron microscopy (SEM, Gemini SEM 300, ZEISS, Oberkochen, Germany) and transmission electron microscopy (TEM, JEM-2100, JEOL, Tokyo, Japan). The X-ray diffraction (XRD) pattern was recorded by an X-ray diffraction analyzer (Ultima IV, Rigaku, Kyoto, Japan) with Cu-Kα radiation (λ_ra_ = 0.15406 nm) to analyze the chemical composition of the NSOC-800. Raman spectroscopy (in Via, Renishaw, London, UK) and X-ray photoelectron spectroscopy (XPS) using a Thermo Scientic K-Alpha (Thermo Fisher, Waltham, MA, USA) were conducted to further characterize the chemical composition. The electrochemical performances of the NSOC-800 electrode were investigated in a three-electrode system with an NSOC-800 as the work electrode, a platinum electrode as the counter electrode and a calomel electrode as the reference electrode in 6 M KOH electrolyte. The electrochemical studies of the NSOC-800//NSOC-800 symmetric supercapacitor were carried out in a two-electrode system with 6 M KOH electrolyte and the same area of NSOC-800 electrodes. The electrochemical tests were measured on a CHI 660E electrochemical workstation, including cyclic voltammetry (CV), galvanostatic charge–discharge (GCD) and electrochemical impedance spectroscopy (EIS, 10^5^ to 10^−2^ Hz). The specific capacitance (F/g) of the working electrode was calculated from the GCD profiles at different current densities.

### 2.4. Computational Process

The specific capacitance, energy density and power density were estimated from the GCD curve with the following equations,
(1)$$C=\frac{It}{\Delta Vm}$$
(2)$$E=\frac{C{\left(\Delta V\right)}^{2}}{2\times 3.6}$$
(3)$$P=\frac{E\times 3.6}{t}$$
where C (F/g), E (Wh/kg) and P (kW/kg) are the specific capacity, energy density and power density, correspondingly. I (A) is the discharge current, t (s) is the discharge time, ΔV (V) is the operating voltage window and m (g) is mass loading of active materials.

## 3. Results and Discussion

### 3.1. Material Characterization

The porous carbon derived from grapefruit peel (denoted as NSOC-800) was synthesized by a facile one-step KOH activation method without extra dopants. Specifically, the homogeneous mixture of grapefruit peel and KOH was annealed in N_2_ atmosphere, and then the KOH was cleaned by the deionized water. The chemical composition and structural characteristics were firstly analyzed by X-ray diffractometry (XRD). As illustrated in Figure 1a, the XRD pattern of the NSOC-800 sample displays two typical peaks at around 23° and 43°, ascribed to the graphitic stacking of (002) and the reflections of the overlapped (100) and (101) faces, correspondingly, which demonstrates that the obtained NSOC-800 sample is a crystalline carbon material [43]. The scanning electron microscopy (SEM) images of the NSOC-800 sample in Figure 1b shows glossy morphology with various sizes. The elemental distribution in the NSOC-800 sample was analyzed by energy-dispersive spectroscopy (EDS) elemental mapping. Figure 1c reveals the presence of C, S, O and N elements in the sample, whose weight proportions are 79.34%, 11.72%, 7.50% and 1.44%, correspondingly. As shown in Figure 1d, the high-resolution transmission electron microscopy (TEM) image displays numerous micropores in the obtained sample, demonstrating its well-developed porosity characteristic. Moreover, the nitrogen adsorption–desorption isotherms of the NSOC-800 sample are revealed in Figure 1e, showing the NSOC-800 sample with a Brunauer–Emmett–Teller (BET) surface of 1267.913 m^2^/g. Additionally, the obtained NSOC-800 sample exhibits micropores of 0.6–2.0 nm diameter and the total volume of 0.189 cm^3^/g, which are favorable for enhancing energy storage (Figure 1f).

The Raman spectra were collected to analyze the graphitization degree of carbon materials in Figure 2a. The characteristic Raman peaks observed at 1350 cm^−1^ and 1590 cm^−1^ are attributed to the D-band (the defect disordered frameworks) and G-band (hybridized carbon architecture (sp^2^)). The relative intensity ratio of the D- and G-bands (denoted as I_D_/I_G_) are negatively correlated with the degree of graphitization. Therefore, the NSOC-800 sample has a comparatively higher I_D_/I_G_ value (1.31), which illustrates that the NSOC-800 derived from grapefruit peel under alkaline conditions endows highly disordered carbon frameworks with abundant defect sites [44]. Furthermore, X-ray photoelectron spectroscopy (XPS) was performed to detect the chemical composition of the obtained sample. The XPS survey of the NSOC-800 sample in Figure 2b shows four elements’ (C, S, N and O) signals, preliminarily notarizing the successful doping of S, N and O in the carbon framework. The high-resolution C 1s spectrum of the NSOC-800 sample, Figure 2c, displays some spin–orbit doublet peaks at 284.4, 285.2, 286.3 and 288.6 eV assigned to C-C, C-N/C-S, C-O and O-C=O. As shown in Figure 2d, the N 1s spectrum reveals three characteristic peaks indexed to the pyridinic-N (397.2 eV), pyrrolic-N (400.0 eV) and graphitic-N (402.6 eV). And, the S 2p spectrum (Figure 2e) shows two peaks at 163.8 and 164.7 eV, which are identical to S 2p_3/2_ and S 2p_1/2_ peaks [45]. Moreover, the two main signal peaks in the high-resolution O 1s spectrum (Figure 2f) suggest the existence of C=O (531.4 eV), C-OH (532.6 eV) and O=C-O (533.9 eV) [42]. The dopant of N, S and O are favorable for the formation of disordered carbon frameworks with abundant defect sites, which corresponds to the Raman results.

### 3.2. Electrochemical Properties

The electrochemical properties of the NSOC-800 sample are conducted in a three-electrode system with 6 M KOH as the aqueous alkaline electrolyte. The cyclic voltammetry (CV) curve was firstly collected at different scan rates in the potential range of −1~0 V. As is visible in Figure 3a, the CV curves of the NSOC-800 exhibit relatively large CV-circulated areas, sustaining quasi-rectangular shapes with a slight distortion, which is ascribed to the combination of the main charge storage behavior of typical electrical double-layer capacitors (EDLC) and the pseudo-capacitance of heteroatoms owing to the incorporation of N, S, O self-doping in the carbon framework [46]. Among them, the rectangular-like shapes of the CV curves can hold up well when the scan rate increases to 100 mV/s, possibly attributed to the distinguished diffusion and electrical conduction of the NSOC-800. Additionally, the excellent energy storage ability of the N, S, O self-doped NSOC-800 electrode is validated by the galvanostatic charge and discharge (GCD) measurement at different current densities (1~10 A/g) in Figure 3b. The GCD profiles of NSOC-800 manifest as isosceles triangle-like shapes, in turn proving the main EDLC behavior of the obtained electrode. The obvious symmetry of the curves suggests that the NSOC-800 endows prominent electrochemical reversibility. The NSOC-800 electrode reveals admirable specific capacitances from 280 to 30 F/g at various current densities from 1 to 10 A/g, which are calculated from the GCD profiles, together with Coulombic efficiencies of 100, 100, 96.4, 93.0 and 90.8%, correspondingly.

To better elucidate the remarkable electrochemical properties of the NSOC-800 electrode, electrochemical impedance spectroscopy (EIS) was carried out in Figure 3c. The Nyquist plot obtained from the EIS includes a semicircle in the high-frequency region and a line with a slope in the low-frequency region, which is related to the charge-transfer resistance and the diffusion-limited electron transfer process [47]. Encouragingly, the NSOC-800 exhibits low charge-transfer resistance (0.58 Ω) and a large slope, confirming that the NSOC-800 with the N, S, O self-doping shows excellent conductivity and ion diffusion rate, which are conducive to the NSOC-800 having an outstanding electrochemical performance. Moreover, the cycling durability is an important performance parameter for electrochemical energy storage devices, and was examined by a GCD test at the current density of 1 A/g. As indicated in Figure 3d, the NSOC-800 presents appealing lifespan with 90.1% capacitance retention after 5000 cycles, demonstrating the potential for practical application.

To further identify the feasibility of the NSOC-800 with N, S, O self-doping in a symmetric supercapacitor device, the symmetric supercapacitor device assembled with NSOC-800 electrodes was fabricated with 6 M KOH electrolyte (denoted as NSOC-800//NSOC-800). As exhibited in Figure 4a, the CV curves of the NSOC-800//NSOC-800 device at various scan rates in the potential window of 0~1.2 V reveals a quasi-rectangular shape, indicating the EDLC behavior and reversible property, which correspond with the GCD curves at different current densities of this device. Figure 4b displays the specific capacitance measured from GCD profiles of the NSOC-800, in which the specific capacitances are 24.3, 22.5, 18.7, 18.1 F/g at current densities from 1 to 8 A/g, together with Coulombic efficiencies of 97.3%, 93.7%, 92.4%, 90.9%, correspondingly. Subsequently, the transport kinetics and ion diffusion process of the NSOC-800//NSOC-800 device were investigated via an EIS measurement. As seen from Figure 4c, the Nyquist plot of the NSOC-800//NSOC-800 exhibits a low charge-transfer resistance of 1.2 Ω and a sloping line with a high slope, further implying that the NSOC-800 electrode has high electronic conductivity and ion diffusion rate. As expected, the NSOC-800//NSOC-800 device presents a conspicuous cyclic performance with only 0.0038% capacitance attenuation of each cycle per 5000 cycles on average (Figure 4d) that is superior to some previously reported works, which are listed in Table 1 [30,42,48,49,50]. Additionally, the NSOC-800//NSOC-800 device delivered a maximum energy density of 5 Wh/kg with a power density of 473 W/kg.

## 4. Conclusions

In summary, a facile and low-cost KOH activation approach without extra dopants is capitalized on the synthesis of N, S, O self-doped carbon with cost-effective grapefruit peel. Profiting from the heteroatoms doping (N, O and S), the NSOC-800 sample exhibits high electronic conductivity and ion diffusion rate. Therefore, a NSOC-800 electrode in alkaline electrolyte delivers a high specific capacitance of 280 F/g at 1 A/g. Moreover, the symmetric supercapacitor device based on the NSOC-800 electrodes exhibits a maximum energy density of 5 Wh/kg with a power density of 473 W/kg. Undoubtedly, such a glorious NSOC-800 electrode with N, O, S self-doping provides a new perspective and insight into designing electrode materials for energy storage devices.

## Figures and Tables

**Figure 1 materials-16-04577-f001:**
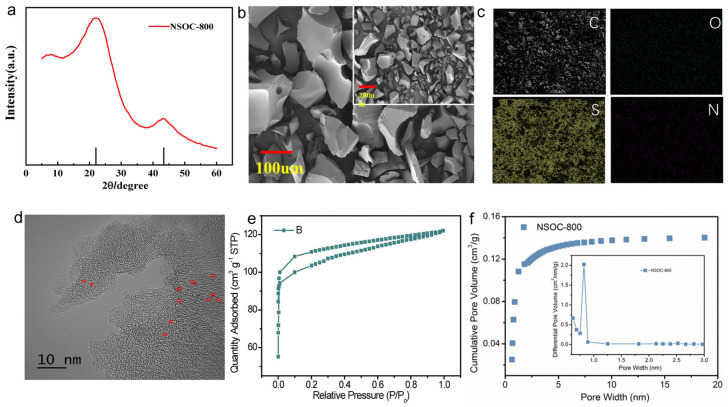
(**a**) XRD pattern, (**b**) SEM image, (**c**) SEM selected-area elemental mapping image and (**d**) high-resolution TEM image, (**e**) Nitrogen adsorption–desorption isotherms and (**f**) Cumulative pore volume and (inset) pore-size distribution of the NSOC-800 samples.

**Figure 2 materials-16-04577-f002:**
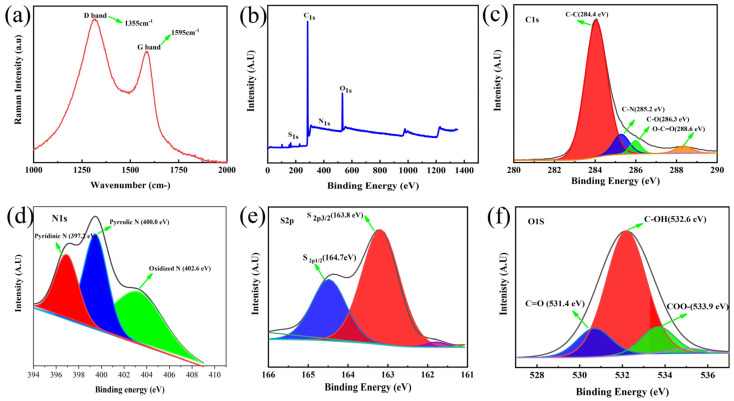
(**a**) Raman spectra, (**b**) XPS survey spectra, (**c**) C 1s XPS spectrum, (**d**) N 1s XPS spectrum, (**e**) S 2p XPS spectrum and (**f**) O 1s XPS spectrum of NSOC-800.

**Figure 3 materials-16-04577-f003:**
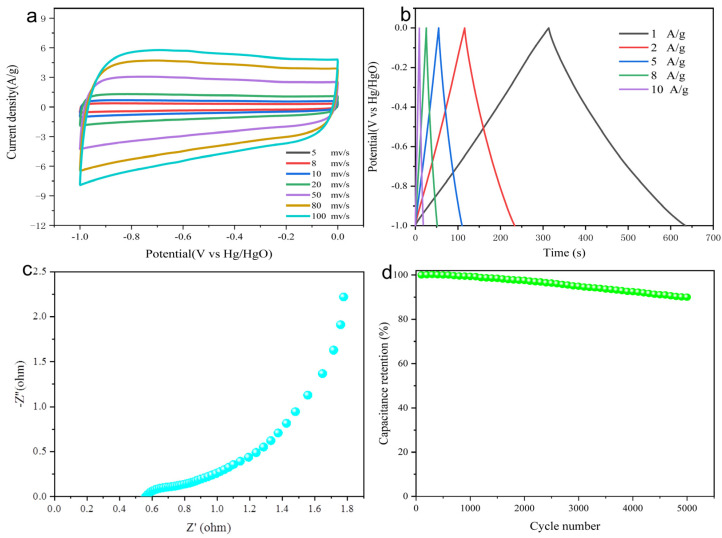
(**a**) CV curves at the scan rates of 5~100 mV/s. (**b**) GCD curves at the current densities of 1~10 A/g, (**c**) Nyquist plots and (**d**) Cycling test at the current density of 1 A/g of NSOC-800.

**Figure 4 materials-16-04577-f004:**
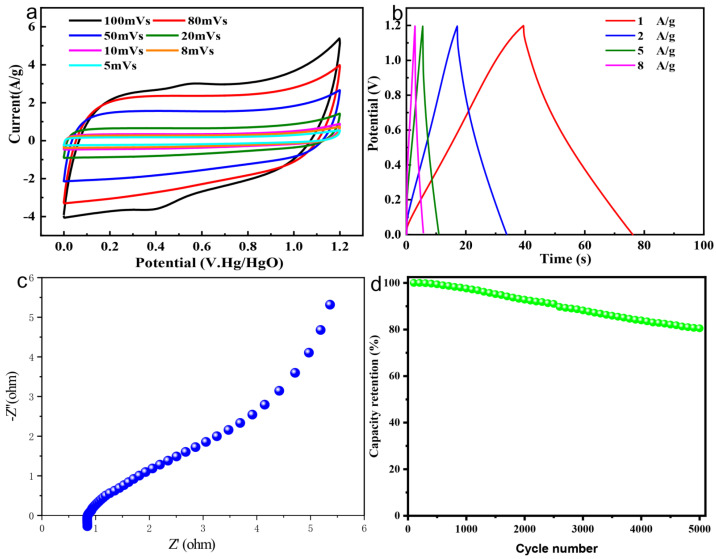
(**a**) CV curves at 5~100 mV/s, (**b**) GCD curves at 1~8 A/g, (**c**) Nyquist plots and (**d**) cycling stability at 1 A/g of NSOC-800//NSOC-800 device.

**Table 1 materials-16-04577-t001:** The lifespan compassion with some reported earlier works.

Supercapacitors	Current Densities (A g^−1^)	Cycles	Capacity Retention (%)
YP-50F//YP-50F (KOH) [48]	0.06	6000	~76.9
N/S-3DGH//N/S-3DGH [42]	2	6000	~76
5 mol L^−1^ LiNO_3_||6 mol L^−1^ [49]	1	5000	~79.5
CLPE based supercapacitors [50]	1	2000	~100
CSC/S//CSC/S [30]	0.335	150	~85
NSOC-800//NSOC-800 (Our work)	1	5000	~81

## Data Availability

Not applicable.

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
