# Peer review of "N, S, O Self-Doped Carbon Derived from Grapefruit Peel for High-Performance Supercapacitors"

_materials, 2023, doi:10.3390/ma16134577_

Round 1

Reviewer 1 Report

The authors have studied the N, S, O Self-doped Carbon Derived from Grapefruit Peel as electrode materials for supercapacitors. Biomass derived carbon are highly attractive for energy storage device, and capacitor studied showed better performance. However, the authors have failed to study the textural and structural properties of the biomass carbon which is scientifically important. I strongly recommend to address the following comments before acceptance:

1)      What is the BET surface area of the carbon? Studying the porosity, pore distribution, pore volume is very important for carbon electrodes in supercapacitors.

2)      Why did SEM images do not show any pores over the surface? The carbon activated with KOH should have combination of macro – meso – micro pores.

3)      The authors should perform TEM to confirm the pore distribution.

4)      What is the electronic conductivity value of the carbon?

5)      Elemental analysis (CHNOS) analysis should be performed to exactly quantify the N, S, and O content in wt%.

6)      Coulombic efficiency should be studied for Fig 3b and 4b.

7)      Give a detailed information on capacitance calculation, and mass loading contents.

8)      Reference section needs an update. Following references regarding biomass carbon should be referred: (i) Energies 2023, 16(2), 802; (ii) Nanomaterials 2020, 10(6), 1220;  (iii) J. Mater. Chem. A, 2018, 6, 17751-17762

N/A

Reviewer 2 Report

In the study titled "N, S, O Self-doped Carbon Derived from Grapefruit Peel for 2 High-performance Supercapacitors", although the presence of N, S, and O in the material content is explained by the XPS result in the method part used, the presence of these elements in graphene should be explained by experiment section.

About conclusion: only the information obtained from the study was given in the conclusion part of the study, no comment was made on a scientific explanation;

About: references: The references used in the study are sufficient for the current study, but there are many supercapacitor studies on graphene, but the references used in the current study, it seems to me too little, no comparison has been made with the previous studies;

About Figures: I did not see any problems with the shapes used.

Reviewer 3 Report

Looking at the symmetric device CV it is certain that, after the potential 1.2 V the surface of Ni-foam getting activated to form Ni(OH)2 on to surface of Ni-foam which showing battery type charge storage broad oxidation and reduction peak. So the estimated charge storage in device is addition of activated carbon and NiOOH form Ni-foam surface. The device should have to operate potential below 1.2 V. Author must re-do the symmetric device electrochemical study properly.

Reviewer 4 Report

Review report:

Authors reported “N, S, O Self-doped Carbon Derived from Grapefruit Peel for High-performance Supercapacitors”. The organization of this work is good, and the discussion is well organized. The characterization and calculation are both solid for the conclusion. Nevertheless, I have some comments which are listed below.

1.     The synthesis scheme (Fig. 2) is not clear, and it should be revised in a detailed way.

2.     The author claimed that their work is a novel investigation, however, myriads of works related to LiNi0.8Co0.1Mn0.1O2 “has been published to date. So, authors should change the way of the presentation focusing on novelty. The introduction should be improved with a paragraph describing the novelty and importance of the work.

3.     The authors must carefully claim their novelty in the INTRODUCTION. In addition, the authors need to do some formatting errors that should be carefully checked and corrected in the text.

4.     The source and purity of all chemicals used should be specified. Authors should be looked at the below-suggested references and can cite and take references regarding the “Source and Purity issues”: “Nanomaterials, 202212(22), 3982”, which references should be cited in your revised manuscript for better understanding.

5.     A summary of key improvements compared to findings in the literature [provide a couple of references to indicate key improvements].

6.     The authors should confirm that in the “Experimental section” in the Electrochemical studies having a “LiPF6 (EC: DEC=1:1) as the electrolyte” electrolyte solution. Here, my point is why the author not used ‘‘Li2SO4, SPVA-HRG, 1 M H2SO4’ and ‘other sources’ instead of ‘KOH’ electrolyte’?

7.     Please provide the comparison table, which you need to prove that your material is superior to previously reported literature.

8.     The authors should add some literature descriptions to make the manuscript more convincing. I would like to suggest the authors cite the following relevant articles to enhance the background; Energies, 11 (2018), pp. 3285”, Nanomaterials, 12 (2022), pp. 3187”

9.     The reviewer also suggests that authors get professional English services to correct the grammatical error and refine the expressions in the body of the manuscript.

10.  Authors should be trimmed/condensed the ‘Abstract’ and ‘Conclusion’ sections in the revised manuscript. Please keep highlights of the whole manuscript in both sections.

Extensive editing of English language required

Round 2

Reviewer 2 Report

Dear author, after the revision you have made, your current study can be published in the journal you have applied.